# An Overview of Alternative Splicing Defects Implicated in Myotonic Dystrophy Type I

**DOI:** 10.3390/genes11091109

**Published:** 2020-09-22

**Authors:** Andrea López-Martínez, Patricia Soblechero-Martín, Laura de-la-Puente-Ovejero, Gisela Nogales-Gadea, Virginia Arechavala-Gomeza

**Affiliations:** 1Neuromuscular Disorders Group, Biocruces Bizkaia Health Research Institute, 48903 Barakaldo, Spain; andrea.lopezmartinez@osakidetza.eus (A.L.-M.); patricia.soblecheromartin@osakidetza.eus (P.S.-M.); delapuente97@gmail.com (L.d.-l.-P.-O.); 2Neuromuscular and Neuropediatric Research Group, Germans Trias i Pujol Research Institute, Campus Can Ruti, Universitat Autònoma de Barcelona, 08916 Badalona, Spain; gnogales@igtp.cat; 3Ikerbasque, Basque Foundation for Science, 48009 Bilbao, Spain

**Keywords:** myotonic dystrophy, spliceopathy, *DMPK*, MBNL, CELF1

## Abstract

Myotonic dystrophy type I (DM1) is the most common form of adult muscular dystrophy, caused by expansion of a CTG triplet repeat in the 3′ untranslated region (3′UTR) of the myotonic dystrophy protein kinase (*DMPK*) gene. The pathological CTG repeats result in protein trapping by expanded transcripts, a decreased *DMPK* translation and the disruption of the chromatin structure, affecting neighboring genes expression. The muscleblind-like (MBNL) and CUG-BP and ETR-3-like factors (CELF) are two families of tissue-specific regulators of developmentally programmed alternative splicing that act as antagonist regulators of several pre-mRNA targets, including troponin 2 (*TNNT2*), insulin receptor (*INSR*), chloride channel 1 (*CLCN1*) and *MBNL2*. Sequestration of MBNL proteins and up-regulation of CELF1 are key to DM1 pathology, inducing a spliceopathy that leads to a developmental remodelling of the transcriptome due to an adult-to-foetal splicing switch, which results in the loss of cell function and viability. Moreover, recent studies indicate that additional pathogenic mechanisms may also contribute to disease pathology, including a misregulation of cellular mRNA translation, localization and stability. This review focuses on the cause and effects of MBNL and CELF1 deregulation in DM1, describing the molecular mechanisms underlying alternative splicing misregulation for a deeper understanding of DM1 complexity. To contribute to this analysis, we have prepared a comprehensive list of transcript alterations involved in DM1 pathogenesis, as well as other deregulated mRNA processing pathways implications.

## 1. Introduction

Myotonic dystrophies are a group of complex, dominantly inherited, disorders characterized by a multisystemic affection. Main alterations include skeletal muscle weakness and wasting, muscle hyperexcitability (myotonia), cataracts, cognitive impairment, gastrointestinal problems, cardiac conduction complications and insulin resistance, amongst other symptoms [1,2]. There are two clinical and molecular forms of myotonic dystrophies described, both inherited in autosomal dominant pattern: myotonic dystrophy type I (DM1) or Steiner’s disease (OMIM 160900), with an incidence of approximately 1 in 8000 people [3,4], and myotonic dystrophy type II or DM2 (OMIM 602668), with an incidence of 1 in 20,000 people [5].

### 1.1. Myotonic Dystrophy Type I

Steiner’s disease is the most common form of adult muscular dystrophy. It is also one of the most variable human disorders with an age of onset ranging from foetal to late-adult age, as well as a wide range of systems affected [6]. Myotonic dystrophy type I (DM1) is caused by a CTG repeat expansion in the 3′ untranslated region (3′UTR) of the DM protein kinase (*DMPK*) gene on chromosome locus 19q3.3 [7,8]. *DMPK* CTG repeats can range from 50 to thousands in DM1 patients [9] and between 5 and 35 in non-DM1 [10] individuals. Interestingly, the number of repeats in blood from DM1 patients is correlated with the disease severity and age of onset, although it might be variable even among family members. In fact, the length of the repeat alone cannot explain the disease variability. In recent years, interruptions at the 5′ and 3′ ends of the CUG tract of pathological *DMPK* transcripts have been described in 3–5% of DM1 patients [11,12]. Although these sequences are mainly composed by unstable CCG interruptions, CGG, CTC and CAG interruptions have also been reported, suggesting a mechanism of phenotypical variability, although further characterization studies are needed [11,12,13,14]. It is important to highlight that inherited mutation length increases during an affected individual lifespan, particularly in differentiated cells. This results in different cells expressing varying repeat lengths [15,16]. The effect of the somatic mosaicism was described when comparing *DMPK* mutant allele repeat expansions in blood cells with skeletal muscle samples, as the latter are much larger: up to thousands in different myonuclei [17,18].

The pathogenic mechanism that leads to DM1 underlying CTG repeats was first associated with *DMPK* haploinsufficiency due to the inability of the protein to complete its final location and function and by the RNA toxicity model, where trapping of proteins within expanded transcripts lead to misregulation of alternative splicing. Nevertheless, abnormal expansion of CTG repeats in *DMPK* might also disrupt the chromatin structure and affect the expression of neighboring genes such as SIX5 and, in some cases of very long repeat expansions, affection of the dystrophia myotonica WD repeat-containing protein (*DMWD*) gene. In a recent work [19], it was suggested that the longer the CTG repeat, the more genes may be affected. DM1 complex symptomatology is indeed derived from a complex pathogenesis, for which further studies are needed [19].

### 1.2. Alternative Splicing

Alternative splicing is a regulatory mechanism of gene expression that contributes to the proteome complexity. It allows the generation of more than one unique mRNA specie from a single gene, and this explains how from as few as 38,268 genes there are as many as 109,005 mRNAs [20]. Splicing is performed in the spliceosome, a large nuclear macromolecular complex that comprises small nuclear ribonucleoprotein particles and many regulating factors, such as RNA-binding proteins (RBPs). RBPs are responsible for the induction of the inclusion or exclusion of different exons in each transcript depending on the specific cell context, cell type and tissue-specific developmental process [21,22]. Transcriptomic studies describe dynamic alternative splicing networks in a range of tissues from adult organs such as brain, heart and skeletal muscle, to embryonic stem and precursor cells, particularly during differentiation or reprogramming of various cell lineages as well as epithelial–mesenchymal transitions. Alternative splicing contributes to cell differentiation and lineage determination, tissue identity acquisition and maintenance and organ development [23,24,25]. RBPs are specific for each differentiation status and contribute to splicing coordination. In fact, genes regulated by alternative splicing are not usually modulated at their overall expression levels (they are typically up- and down-regulated) [22,26].

Skeletal muscle is one of the tissues with the highest number of differentially expressed exons. Its splicing program is so complex that it might differ even to that of cardiac muscle. In skeletal muscle, the muscleblind-like (MBNL) family, CUG-BP and ETR-3-like factors (CELF) family and the RNA binding Fox (RBFOX) are the most important splicing regulators.

Within expanded *DMPK* transcripts, CUG repeats form imperfect stable hairpin structures that accumulate in the cell nucleus in small ribonuclear complexes or microscopically visible inclusions, which impair the physiological function of proteins implicated in transcription, splicing or RNA export, whereas it is not clear how many proteins other than MBNL are trapped in the ribonuclear foci [27,28]. These aggregations lead to the deregulation of the alternative splicing of different transcripts due to the alteration of the splicing machinery, specifically *MBNL1* and CELF1. Hundreds of splicing events are misregulated due to *MBNL1* sequestration and CELF1 upregulation [29,30]. These alterations in turn cause loss of cell function and viability, and some of them directly correlate with common symptoms of the disease, explaining its extensive multisystemic affection [31,32].

In this review we focus on *MBNL1* and CELF1 implication in alternative splicing to help understand DM1 pathogenic alterations caused by this deregulation. To contribute on this analysis, we have prepared a comprehensive list of transcript alterations involved in DM1 pathogenesis. 

## 2. Muscleblind-Like (MBNL) Proteins and Mantaining MBNL/CELF1 Equilibrium in DM1

### 2.1. MBNL-Dependent Splicing Regulation

Muscleblind-like proteins (MBNL) are a family of tissue-specific regulators of developmentally programmed alternative splicing. They regulate alternative splicing events including cassette exon, 5′ splice site, 3′ splice site, mutually exclusive exons, intron retention and alternative 3′ end formation by alternative cleave and polyadenylation [33,34]. Some studies also propose a role in translational control through a modulation of RNA stability, mRNA localization and microRNA biogenesis [35]. In mammals, MBNLs are coded by the genes *MBNL1*, expressed in most tissues, *MBNL2*, mainly expressed in brain but also in skeletal muscle and *MBNL3*, whose expression is more restricted to muscle cell differentiation and regeneration situations, as well as in placenta [36,37,38,39].

All MBNLs share structural similarities, as represented in Figure 1. The three MBNL paralogs are composed of different exon combinations and may also be alternatively spliced. *MBNL* exons 1, 2 and 4 encode for four zinc fingers (ZnF) domains arranged in two similarly structured tandem pairs that are important for RNA binding and splicing activity. Exon 3 encodes for a linker sequence essential for RNA/protein interactions as it is hypothesized to increase protein flexibility, allowing the binding of the ZnF to a wide range of targets with different structures [40,41]. Exon 5 and the KRAEK motif from exon 6 encode for a bipartite nuclear localization signal (NLS) and exon 7 is important in the formation of homotypic interactions, favouring MBNL–MBNL interactions when target motifs are nearby [42,43,44,45].

*MBNL1* gene codes for 10 exons, marked as 1 to 10 in Figure 1, even though not all of them are included in all *MBNL1* transcripts. Six of the coding exons are alternatively spliced (3, 5, 6, 7, 8 and 9) depending on the tissue and the developmental point in which *MBNL1* is expressed. *MBNL1* transcript variants with exons 5 and 7 are mainly described in early differentiation stages and in adult DM1 tissues. These exons enhance sequestration of MBNLs in nuclei of DM1 cells and thus contribute to the severity of the phenotype by favouring MBNLs homotypic interactions promoted by the tandem arrayed *DMPK* triplet expansion [41,46,47].

MBNLs interact with RNA targets via ZnF domains that bind 5′-YGCY-3′ motifs where Y are pyrimidines [43]. These motifs are abundant in CUG expanded RNAs but also in CAG and CCG repeats [43,48]. MBNLs bind the same consensus motif in introns and 3′UTRs, indicating the same mode of interaction with various RNA structural segments and in different cellular compartments. The RNA processing activity of MBNL proteins is modulated by both the number and structural context of these binding motifs [49]. Furthermore, MBNL binding to exons and introns is part of the regulation of the pre-mRNA splicing. Its binding affinity to recognised sequences within target pre-mRNAs correlate with its splicing activity. The ZnF binding site relative to the regulated exon also defines MBNL splicing function: binding within the alternative exon and upstream intronic regions generally facilitates exon skipping while binding to downstream intronic regions promotes exon inclusion [33,50]. Meanwhile binding to 3′UTRs is linked to interactions with mature mRNAs for mRNA stability and cellular localization, supporting translation and protein secretion pathways [33,43,51].

Muscle fibres formed during prenatal development are extensively remodelled after birth. Postnatal remodelling may require activation of transcriptional programmes that are not initially induced during muscle differentiation, where alternative splicing plays a key role. *MBNL1* is essential in the postnatal remodelling of skeletal muscle by promoting two key developmental transitions: the promotion of the differentiation of embryonic stem cells and the induction of a shift from a foetal to an adult splice pattern of target RNAs. Likewise, *MBNL1* downregulation causes an adult-to-foetal alternative splicing transition. Indeed, neonatal mice with depleted *MBNL1* expression were unable to perform the transcriptomic transition between foetal and post-natal patterns [50,52,53,54]. On the other hand, *MBNL2* acts mainly during earlier stages of muscle development, even though in mature tissues it is located in the cytoplasm regulating mRNA decay. Nevertheless, a compensatory role between *MBNL1* and *MBNL2* in mature tissues has been described as *MBNL2* expression increases following functional loss of *MBNL1* [55,56].

Studies in mouse skeletal muscle suggest that *MBNL1* has a predominantly cytoplasmic location during early stages of neonatal development, while at the end of the process is predominantly nuclear, coinciding with the postnatal splicing transitions (Figure 2). Therefore, postnatal splicing transitions are triggered, at least partly, by translocation of *MBNL1* from the cytoplasm to the nucleus. Nevertheless, *MBNL1* is not completely depleted in the cytoplasm of muscle fibres of neonatal mice [52]. In contrast, during postnatal development in cardiac muscle, *MBNL1* concentration is increased 4-fold [53].

As represented in Figure 2, while *MBNL1* nuclear levels increase during development, CUG-BP and ETR-3-like factor 1 (CELF1) nuclear levels decrease. CELF1 binds to single-stranded UG-rich RNA sequences but not to double-stranded sequences, unlike *MBNL1*, suggesting a key difference in their splicing regulatory mechanisms. *MBNL1* and CELF1 act as antagonist regulators of several pre-mRNA targets, including cardiac troponin (cTNT), insulin receptor (*INSR*), chloride channel 1 (*CLCN1*) and *MBNL2* [53,54,57].

### 2.2. MBNL and CELF1 Implication in DM1

MBNL family proteins preferentially recognize CUG repeats when they are pathologically expanded. Due to the weak pairing of U–U bases, the trinucleotide repeat sequences form a secondary structure with unpaired mismatches that make RNA strands accessible for MBNL ZnFs interaction with the GC motifs [58,59]. MBNL activity is controlled by RNA secondary structures; it preferentially binds single-stranded RNAs rather than more stable RNA structures. Indeed, base pairing of YGCY motifs or their structural stabilization was shown to impair splicing. In Figure 3, MBNL binding to weak paired YGCY motifs is represented [49,60].

A recent study showed that one single *MBNL1* protein binds a (CUG)4 while four *MBNL1* particles interact with (CUG)12 [61,62]. A low level of expanded CUG RNA is readily saturated with MBNL proteins, which dynamically exchange with the unbound MBNL units in the nucleoplasm, while high CUG repeat lengths severely deplete this pool. *MBNL1* binds pre-mRNA targets and CUG repeats with similar affinity, suggesting that the consequent toxicity is directly related to the length of expansion and the number of MBNL units trapped in the ribonuclear foci [63]. Nevertheless, even though the main components of the ribonuclear foci are MBNL and *DMPK* transcripts, additional proteins from the spliceosome have been proposed to be sequestered within the triplet repeats, which can also influence the disease pathogenesis [63,64]. MBNL retention into the nucleus by the *DMPK* CUG repeats consequently decreases *DMPK* and MBNL functional concentrations in cytoplasm, as *DMPK* mRNA is withheld in the nucleus and translation is impaired [58,65]. In fact, CUG expansion size progressively increases in mature skeletal muscle, leading to the complete depletion of *DMPK* protein and free *MBNL1* in the nucleoplasm. The *DMPK* loss is also sometimes described as *DMPK* haploinsufficiency, which might also have implications in DM1 molecular mechanisms, as well as the alterations in the mRNA transport and stability and in the miRNA synthesis derived from the protein trapping in ribonuclear foci [33,66]. The splicing defects described in DM1 are strikingly similar to those observed in *MBNL1* knockout mice but not in *MBNL2* defective mice, leading to the conclusion that *MBNL1* has a pivotal role in DM1 pathogenesis, independently of *MBNL2* [52]. Nevertheless, *MBNL2* also plays an important role in the splicing defects described in brain tissues [44] and indeed, MBNL paralogs contribute to the misregulation of alternative splicing through an additive effect in different tissues [30,67].

*MBNL1* sequestration is enough to explain misregulated splicing in adult DM1 skeletal muscle, so just one splicing factor missing is responsible of most downstream effects. Nevertheless, misregulation of CELF1 in DM1 has also been extensively studied in the last few years: although alterations of CELF1 distribution or amount are not required to produce splicing regulation defects in DM1, it contributes to the induction of the embryonic splicing patterns (Figure 3) [52,68]. As mentioned before, *MBNL1* and CELF1 act as antagonistic regulators of alternative splicing so the loss of functional *MBNL1* induces CELF1 upregulation. Although with lower affinity than *MBNL1*, CELF1 binds to CUG-repeat containing RNA [69,70,71]. CELF1 steady-state levels are increased in DM1 through the activation of the protein kinase C (PKC) signalling pathway that promotes CELF1 hyperphosphorylation and stabilization, while its relation with the binding to CUG repeats remains unclear [68,71]. Unlike *MBNL1*, which binds to the hairpin structures, CELF1 does not co-localize with the CUG expansion RNA in the nuclear foci, as it preferentially binds to single-stranded CUG repeats. That means that CELF1 steady-state protein levels are increased in DM1 patients while *MBNL1* levels are just reduced to the nuclear foci [68,71]. Since CELF1 promotes foetal alternative splicing events, its overexpression plays a key role in the inversion to foetal splicing patterns in adult tissue, the main characteristic of the spliceopathy (Figure 3) observed in DM1 [52,53,54].

## 3. Spliceopathy Due to RNA Toxicity

Alternative splicing events are generally induced by cis-acting regulatory elements within pre-mRNAs that promote or inhibit exon recognition, as well as by the expression or activity of trans-acting factors such as MBNL and CELF proteins. MBNLs and CELFs bind to these cis elements and regulate the accessibility of the spliceosome to splice sites [53,72]. As represented in Figure 4, mutant *DMPK* mRNAs are spliced and polyadenylated but their nuclear sequestration, due to the CUG repeats in the 3′UTR, prevents translation [73]. The RNA “gain-of-function” by the expanded transcripts causes the accumulation and deregulation of RBPs that act as splicing regulators, such as *MBNL1* and CELF1. This in turn causes a spliceopathy, which is the general alteration of the mRNA processing pathways. Different transcripts from different tissues are incorrectly spliced, causing most of the DM1 symptoms, as reflected in Figure 4. Figure 4 summarises some DM1 symptoms linked, or hypothesized to be related to, the splicing alteration of different genes in pathogenic conditions. Transcript alterations of the genes represented are detailed later in this manuscript.

As well as altering MBNL/CELF1 balance, toxic transcripts also cause the accumulation in the nucleus of splicing and cleavage factors, such as heterogeneous nuclear ribonucleoproteins (hnRNPs) and small nuclear ribonucleoproteins (snRNPs). These factors are needed for the pre-mRNA to generate a mature transcript in the spliceosome and their deregulation can lead to an overall impairment of post-transcriptional pre-mRNA pathways [74]. Therefore, the accumulation of toxic RNAs impedes the functionality of the splicing machinery and reduces the nucleus export and import trafficking, thus impairing mRNA transcription, post-transcriptional modifications, localization, stability and translation [28,75].

It should be noted that altered expression of splicing factors and alternative splicing changes may also happen during active regeneration processes in degenerative muscle diseases [76]. However, in DM1 muscles no massive degeneration/regeneration is observed, while altered splicing events were also found in DM1 non-regenerating tissue such as cardiac tissue [33,77,78]. Nevertheless, the splicing changes described in the following sections might not be unique to DM1, except in those cases where it is specifically mentioned. Recent studies showed that most expression and splicing changes described in DM1 are indeed described in DM2 or other neuromuscular disorders [76,79]. 

### 3.1. Misregulation of mRNA Processing

The misregulation of the alternative splicing is one of the best-characterized effects in DM1 cells. To date, more than thirty transcripts missplicing have been characterized in different tissues in DM1 patients (Figure 4) and more than sixty in mice tissues [53,80]. Misregulated splicing events in DM1 are usually developmentally regulated and exhibit an adult-to-embryonic switch in the splicing patterns, due to *MBNL1* and CELF1 implications in developmental transcription regulation. Some of the altered transcripts fail to meet the adult tissue requirements and thus directly contribute to the overall disease pathology [81]. The extent of symptoms varies depending on the tissue context, such as relative concentrations of MBNL paralogues (*MBNL1*, *MBNL2* and *MBNL3*) and the degree to which they are sequestered [67].

#### 3.1.1. Transcripts Altered in DM1 Brain Tissue

*MBNL2* sequestration in brain tissue in DM1 patients has major consequences, as its expression is higher there than *MBNL1*′s. Indeed, *MBNL2* knockout mice exhibit a number of DM-related central nervous system abnormalities, including irregular REM sleep propensity and deficits in spatial memory [44]. A summary of all the transcripts altered in DM1 brain tissue described in this section is showed in Table 1. 

One of the better-described missplicing due to loss of *MBNL2* is microtubule-associated protein tau (*MAPT*), as found in DM1 frontal cortex samples [82]. Abnormal expression of *MAPT* isoforms, excluding exons 2, 3 and 10, and progressive appearance of neurofibrillary tangles (NFTs) composed of intraneuronal aggregates of hyperphosphorylated tau protein, are present in DM1 patient samples, suggesting a tautopathy-like degeneration of brain tissue [83,84].

NMDA receptor 1 (*NMDAR1*) splicing is also altered in DM1 brain samples. *NMDAR1* function is required for normal long-term potentiation in the hippocampus, contributing to learning processes. Exon 5 inclusion in *NMDAR1* influences the pharmacologic behaviour, gating and cellular distribution (somatic rather than somato-dendritic expression). It is hypothesized that the missplicing alteration may contribute to memory impairment observed in DM1 [85].

The amyloid β precursor protein (*APP*) is also misspliced in DM1 as described in DM1 brain samples. CELF1 promotes the exclusion of exon 7, which encodes for a protease inhibitor domain, and the recapitulation to the foetal isoform of the protein. Nevertheless, no pathological features have been related to this missplicing event [82,85].

The splicing of *MBNL2* and *MBNL1* transcripts itself is also affected in DM1 brain tissue samples, as well as in DM1 skeletal muscle. In adult and DM1 brain tissue, different *MBNL1* and *MBNL2* isoforms are expressed, with an increase of isoforms including exons 7 and 8 in *MBNL2* mRNA and in *MBNL1* mRNA exons 6 and 8 [29,86,87].

#### 3.1.2. Transcripts Altered in DM1 Skeletal Muscle

*MBNL1* is the main alternative splicing regulator in skeletal muscle, working co-ordinately with CELF1. Typically, *MBNL1* translocates from cytoplasm to the nucleus in the postnatal period to induce adult-type splicing, but in DM1 it is depleted from the nucleoplasm because it is recruited extensively into the ribonuclear foci. This leads to the misregulation of alternative splicing of multiple transcripts represented in Table 2, included *MBNL1* itself: exon 7 is developmentally regulated by *MBNL1* homotypic interactions during postnatal development [29,52,86] and it has been described as the inclusion of exons 5, 7 and 10 of the *MBNL1* transcript in DM1 skeletal muscle samples.

*MBNL1*’s own splicing alterations affects other genes by generating different protein isoforms, ablating protein synthesis or changing protein localization, such as ectopic expression of proteins as dystrobrevin α [88] and pyruvate kinase M2 [89]. 

Dystrobrevin-α (*DTNA*) belongs to the dystrobrevin subfamily of dystrophin proteins. It is a component of the dystrophin-associated protein complex, whose main role is to maintain the sarcolemma stability. Mutations in the coding gene are associated with congenital heart defects. Even though its main role has been described in the dystrophin-associated protein complex in muscle, *DTNA* is mainly expressed in the brain and in less proportion, in cardiac and skeletal muscle [90]. In the brain *DTNA* isoform 2 is located in the nucleus and Cajal bodies of neurons. Nevertheless, alterations of splicing in *DTNA* have only been described in cardiac and skeletal muscle [86] and no alterations in brain transcripts related with DM1 have been defined yet [91]. In DM1, misregulation of *DTNA* splicing has been related to muscle weakness due to the ectopic expression of the non-muscular transcript with exclusion of exons 11a and 12, as described in DM1 skeletal muscle samples [29]. *DTNA* is subjected to extensive splicing regulation through the alternative inclusion/exclusion of exons 11b, 17b and 21, leading to different isoforms that are differently distributed in muscle. Isoform 1, which contains exons 1 to 21, is mainly located at the neuromuscular synapse while isoform 2, which lacks a COOH-terminal section as is coded by exons 1 to 17b, is located in the neuromuscular junction [88,92]. 

Pyruvate kinase M2 (PKM2) is a critical enzyme of the glycolytic pathway mainly expressed in the adult brain and kidneys. It is expressed in skeletal and cardiac muscle during embryonic development but suffers a splicing switch: exons 9 and 10 are excluded (PKM1) early in postnatal stages. Although PKM2 is substituted by PKM1 in postnatal heart and skeletal muscle, it is maintained in the brain. PKM2 has been described to be a key enzyme in the Warburg effect in cancer and its splicing is misregulated in DM1. CELF1 and, in a small percentage, *MBNL1* and *MBNL2* regulate PKM2 alternative splicing, causing its ectopic expression in DM1′s heart and skeletal muscle type I fibres, which undergo atrophy in DM1. PKM2 upregulation increases glucose consumption with reduced oxidative metabolism in DM1 cell cultures, suggesting defects in energy metabolism that could foster muscle wasting [89].

Regarding transcripts physiologically expressed in skeletal muscle, different muscle features are altered due to its missplicing. The sarcomere excitation–contraction coupling is a cooperative process in which different proteins are implicated. One of the best-described splicing alterations is the inclusion of exon 7a in the chloride voltage-gated channel 1 (*CLCN1*) transcript. *CLCN1* is the main chloride channel in muscle, responsible for ion conductance and excitability. Its alternative splicing is regulated by *MBNL1*, but not CELF1, and its depletion causes the in-frame inclusion of intron 2 and exon 7a, resulting in the creation of a premature stop codon. The truncated protein is unable to locate in the surface membrane, producing a chloride channelopathy that ends up in membrane hyperexcitability, responsible for myotonia in DM1. Although it is highly expressed in the brain as well, no splicing alterations or implications have been described [93,94,95].

The bridging integrator 1 or amphiphysin 2 (*BIN1*) is a protein involved in tubular invaginations of membranes and is required for the biogenesis of muscle T tubules, which are specialized skeletal muscle membrane structures essential for excitation–contraction coupling. Alternative splicing of *BIN1* pre-mRNA is controlled by *MBNL1* binding so *MBNL1* depletion causes the translation of an inactive form of *BIN1* lacking exon 11. *BIN1* transcript lacking exon 11 has been considered a splicing aberration unique in DM1 as it has not been described in any other pathological or control tissue studied, and it is not corresponding to any other isoform coded in human tissues. The inactive form of *BIN1* impairs the excitation–contraction coupling the sarcomere, which contributes to muscle weakness present in DM1 [29,96].

Deeper in the sarcomere excitation–contraction coupling elements, alterations of the intracellular calcium homeostasis play a key role in muscle degeneration in DM1 [97]. The calcium channel CaV1.1, coded by *CACNA1S* gene, is a voltage-sensitive channel that plays a central role in excitation–contraction coupling. Skipping of exon 29 of the *CACNA1S* transcript is developmentally regulated by *MBNL1* and CELF1 and has been described in other neuromuscular diseases as facioescapulohumeral muscular dystrophy (FSHD), but not to the extent described in DM1 [97]. Skipping of *CACNA1S* exon 29 has been linked with muscle weakness severity [29]. It causes an increased CaV1.1 conductance and voltage sensitivity, raising transient calcium influx, which causes contraction impairment and muscle weakness [97].

Another protein involved in calcium homeostasis with described alternative splicing alterations in DM1 is the ryanodine receptor 1 or *RYR1* (*RYR1*). During a normal skeletal muscle contraction/relaxation cycle, calcium is released from the sarcoplasmic reticulum into the cytoplasm through *RYR1*, inducing muscle contraction. The foetal *RYR1*, lacking exon 70, is increased in DM1 tissues compared with control samples [29,98] and decreases the probability of the channel being open, which subsequently reduces contraction strength by diminishing calcium influx to the cytoplasm.

SERCA1 channels pump back calcium to the lumen of the sarcoplasmic reticulum, allowing skeletal muscle relaxation. Not surprisingly, alternative splicing of the gene encoding SERCA1 (*ATP2A1*), regulated by *MBNL1*, is also altered in DM1. Similar to *RYR1*, SERCA1b, the neonatal form, lacking exon 22, is exclusively expressed in several DM1 muscle samples, together with minor alternatively expressed variants. In non-DM muscles only SERCA1a, the adult form including exon 22, was found. This contributes to the alteration of the intracellular calcium homeostasis and muscle degeneration in DM1 [29,98].

Sarcomere structure is also altered in DM1. Different proteins are involved in the maintenance and mechanical support of the sarcomere during contraction as fast troponin T3, coded by the *TNNT3* gene. In DM1, the inclusion of the foetal exon 23 is promoted by CELF1, although in healthy conditions the alternative splicing of *TNNT3* is regulated by both *MBNL1*-CELF1 [52,99].

A key protein in the maintenance of the muscle structure is dystrophin, coded by the *DMD* gene. In-frame mutations in DMD cause Becker’s muscular dystrophy while out-of-frame mutations cause Duchenne muscular dystrophy (DMD). In DM1, exons 71 and 78 of the DMD gene are excluded from the transcripts. No specific effects have been related to exon 71 omission, but a mild case of DMD due to exon 78 deletion was described, inducing instability of the sarcolemma. This led to the conclusion that missplicing of exon 78 of the *DMD* transcripts in DM1 may contribute to muscle weakness by decreasing membrane integrity [86,100]. 

Splicing of the *CAPN3* gene, coding for the intracellular protease Calpain 3, is also altered in DM1. CAPN3 is mainly expressed in skeletal muscle and is responsible for the cleavage of a big range of proteins implicated in the sarcolemma structure. Mutations in *CAPN3* are responsible for the subtype R1 of limb-girdle muscular dystrophy (LGMD). LGMDR1 or LGMD2A causes progressive muscle weakness in lower and upper limbs with wide progression variability among patients. Different *CAPN3* mutations have been described as causing LGMDR1 but none of these mutations are similar to *CAPN3* splicing alteration described in DM1 samples. *CAPN3* exon 16 exclusion is a unique alteration described just in DM1 samples, which correlates with muscle weakness due to a decrease in protease activity [52,86].

*MYOM1* gene also encodes for a structural component of the sarcomere: myomesin 1, a protein of the sarcomeric M band. The inclusion of *MYOM1* exon 17a has been described in skeletal muscle samples of DM1 patients. The splicing alteration causes instability of myomesin 1, leading to fragility of the sarcomeric M-band that may contribute to muscle weakness. It is also expressed in cardiac muscle, although no splicing alterations of this gene have been described in this tissue [101].

Although it is not exactly a sarcomeric protein, nebulin, coded by the *NEB* gene, is a giant component of the cytoskeleton that provides structural resistance to the myotubes. It is implicated in myotube stability by linking the thick and thin filaments within the sarcomeres of skeletal muscle. Indeed, nebulin accounts for 3–4% of the total myofibrillar protein in most vertebrates. nebulin isoforms are regulated by alternative splicing following tissue and developmental stage-specific patterns. Mutations in *NEB* are associated with recessive nemaline myopathy, the most common non-dystrophic neuromuscular disorder, which also exhibits muscle weakness, but in this case to a higher extent, impairing initial motor development. The inclusion of exon 116 in *NEB* transcripts of DM1 skeletal muscle tissue has been described, but no studies have linked any DM1 pathological feature with this missplicing alteration [86,102].

Similar to nebulin, splicing of nebulin-related anchoring protein (*NRAP*) is altered in DM1. *NRAP* is a cytoskeleton component that plays a mediator role between nebulin and myofibrilar proteins. The extent of exon 12 exclusion in DM1 skeletal muscle samples has been related to alterations of the myofibril assembly. Nevertheless, this transcript excluding exon 12 has also been described in normal skeletal muscle samples, although at a different rate than in DM1 [52,86].

Regarding sarcomere signalling, two *MBNL1*-dependent splicing alterations have been described: the exclusion of exon 11 in *INSR* and exons 2.1, 2.2 and 2.3 in *MTMR1*.

The skeletal muscle insulin receptor (*INSR*) isoform is regulated by a combination of *MBNL1* and *MBNL2*. Deregulation of their splicing activity in skeletal muscle causes the exclusion of exon 11 from the *INSR* transcript, and the switch from the isoform expressed in muscle, *ISNR-B*, into the *INSR-A* isoform, usually expressed in the brain, spleen and leukocytes. This switch in DM1 is not a secondary action to the dystrophic changes or regeneration as it is not observed in other myopathies such as DMD, LGMD or FSHD that impair muscle wasting and aberrant degeneration/regeneration cycles. *INSR-A* codes for a lower-response insulin receptor, which causes a decreased metabolic response to insulin that leads to an unusual form of insulin resistance in DM1 patients [29,103].

Myotubularin-related protein 1 (MTMR1) is a phosphatase involved in muscle formation. *MTMR1* alternative splicing is regulated during myogenesis, involving three coding exons that in combination, code for three different molecular weight proteins. It has been described as the inclusion of exons 2.1, 2.2 and 2.3 in the *MTMR1* transcript in DM1 skeletal muscle samples. All three exons are derived from intron 2 sections and the 2.2 section is a newly coding exon “unique” in DM1. Protein variants including exon 2.2 have not been described in controls, neither in other neuromuscular disorders samples. Although no specific implications in DM1 pathology have been related to this alteration, mice studies have showed impaired myogenesis due to MTMR1 altered splicing [86,104].

Another group of genes with splicing alterations described in DM1 skeletal muscle samples are those coding for extracellular matrix (ECM) components. Experimental data in mice suggest that splicing alterations of genes coding for ECM components as *SMYD1* and *NFIX* may not be caused by *MBNL1*, but likely by *MBNL2* [50]. SET and MYND domain containing 1 (*SMYD1*) is a methyltransferase that acts as a transcriptional repressor in cardiac and skeletal muscle. Indeed, it is essential for cardiomyocyte differentiation and cardiac morphogenesis. In DM1 skeletal muscle samples the inclusion of exon 39 has been described, although no consequences of this altered splicing had been observed in neither cardiac nor skeletal muscle [50]. The nuclear factor IX (NFIX) is a transcription factor involved in extracellular matrix remodelling during myogenesis. The inclusion of exon 7 in skeletal muscle samples of DM1 patients has been reported in as many as 68% of transcripts in DM1 vs. healthy samples and this has promoted its use as an alternative splicing marker in many studies. Nevertheless, no pathological complications have been related to this misregulation [29,86,105]. Lastly, the gene coding for the matrix remodelling associated 7 (*MXRA7*) has also been described as misspliced in DM1. Exclusion of exon 4 was reported in DM1 skeletal muscle samples. *MXRA7* is expressed in a wide range of tissues, and especially in large amounts in prostate samples. No further pathological implications have been described [86].

Although no other studies have been reported about mitochondrial function in DM1 skeletal muscle samples, an alternative inclusion of exon 1 of the *ATP5MC2* gene has been described. This gene codes for an ATP synthase membrane subunit C locus 2 enrolled in the oxidative phosphorylation and no symptoms related to this splicing alteration have been described yet [86].

Finally, there are several other transcripts that report alternative splicing in DM1, without described phenotypical effects. Although described in DM1 skeletal muscle samples, its expression might not be restricted to this tissue:

Son of sevenless homolog 1 (*SOS1*) is a guanine nucleotide exchange factor (GEF), which interacts with Ras proteins to turn guanosine diphosphate (GDP) into guanosine triphosphate (GTP). It is implicated in RAS/MAPK signalling pathways, regulating cell cycle entry and proliferation. It is widely expressed among different tissues, even though it has a characteristic splicing switch in DM1 skeletal muscle samples. Exon 25 exclusion in DM1 samples is the largest alternative splicing effect described, being 99% of exon 25 inclusion in healthy subjects versus the 16% in DM1 subjects. The alternative spliced transcript in DM1 codes for a *SOS1* protein with decreased activity that inhibits signalling pathways involved in muscle hypertrophy [29,86].

The formin homology 2 domain containing 1 (*FHOD1*) protein contributes to actin organization in skeletal muscle. The exclusion of exon 11a has been described in DM1 tissues compared to healthy controls although no alterations in actin organization have been demonstrated in DM1 and it’s not linked with muscle weakness [86]. 

Glutamine-fructose-g-phosphate transaminase 1 or GFPT1 is an enzyme of the hexosamine pathway that controls by protein glycosylation the glucose influx into the pathway. *GFPT* is widely expressed in different human tissues but it is incorrectly spliced in DM1 skeletal muscle samples. In DM1, exon 9 exclusion recapitulates the neonatal isoform of the GFPT1 protein, which has been described as reducing the feedback inhibition of the glucose influx; although no clear cell consequences neither pathological implications have been related to this change [29].

Exon 2 of *ALPK3* gene is included in some transcripts, as described in DM1 skeletal muscle samples. Nevertheless, *ALPK3* is mainly expressed in cardiac muscle, coding for α kinase 3, a protein implicated in myogenesis and cardiac signalling. No cardiac complications are described as derived from this alteration nor other pathological implications [29]. 

The last splicing alteration described in skeletal muscle is the inclusion of exon 10 of the nuclear receptor corepressor 2 (*NCOR2*) in DM1 samples. *NCOR2* is highly expressed in skeletal muscle although no pathological implications have been linked with its missplicing [86].

#### 3.1.3. Transcripts Altered in DM1 Cardiac Muscle

The molecular mechanisms underlying cardiac defects, which affect 80% of individuals with DM1 and represent the second most common cause of death from this disease [108,109], are yet to be defined. Cardiac involvement in DM1 is characterized by a cardiac-conduction delay that may result in fatal atrio-ventricular block, and by atrial or ventricular tachycardia. Similarly to skeletal muscle, *MBNL1* and CELF1 are the main splicing regulators in the heart although in collaboration with other splicing regulators, such as RBFOX1/2 [110]. A summary of all the transcripts altered in DM1 cardiac tissue described in this section is showed in Table 3.

One of the best-described transcript alterations in cardiac muscle is *SCN5*. *SCN5* gene is coding for a α- subunit of the cardiac voltage channel NaV1.5 and its splicing is regulated by *MBNL1*. The sodium channel plays a key role in the excitability of cardiomyocytes due to the rapid propagation of the impulse through the cardiac-conduction system. Mutations described in *SCN5A* lead to a variety of arrhythmic disorders. In DM1 heart samples, exon 6a (foetal cardiac exon) is included in the adult transcript. The conductance of the channel is affected, leading to a slower upstroke velocity of the cardiac action potential that ends up in a slowing of the normal conduction [110].

Another splicing alteration regarding ion channels in heart is SERCA2, coded by *ATP2A2* gene. In contrast with the skeletal muscle variant (SERCA1) whose splicing is developmentally regulated, splicing of SERCA2 is regulated in a tissue-specific manner. It is mainly expressed in cardiac muscle and in less proportion in type I fibre in skeletal muscle. In DM1 skeletal muscle samples, intron 19 inclusion has been described. SERCA2 being the main cardiac isoform and following experimental data in mice models, it has been hypothesized that SERCA2 splicing alteration contributes to the calcium influx dysregulation in cardiac muscle, leading to complications in cardiac conduction [98,111].

Troponin 2, the cardiac isoform of troponin encoded by the *TNNT2* gene, is also implicated in DM1 pathology through the alteration of calcium sensitivity of the muscle fibre. Inclusion of exon 5 of *TNNT2* has been described in transcripts of cardiac tissue samples of adult DM1 patients, as well as in other neuromuscular diseases. This alternative transcript produces the foetal isoform of the protein and confers different calcium sensitivity to the myofilament, affecting the contractile properties of mature muscle. This process is regulated by *MBNL1* in physiological conditions, switching to CELF1 control when *MBNL1* is not functional. Although molecular mechanisms responsible for cardiac defects in DM1 are still unclear, the *TNNT2* incorrect splicing might contribute to the reduced myocardial function and conduction abnormalities seen in DM1 patients [54,111].

Regarding cardiac transcripts encoding for sarcomere structural proteins, the LIM domin binding 3 (*LDB3*) gene or *ZASP* transcript is also misspliced in DM1. LDB3 is a protein located in the Z-line that interacts with actininin, providing structural support to the Z-line during muscle contraction. In DM1, the inclusion of exons 5 (foetal heart isoform) and 11 have been described and linked to morphological abnormalities in cardiac fibres. Misspliced LDB3 is unable to bind protein kinase C (PKC) with enough affinity, which was previously showed to cause dilated cardiomyopathy [29,112].

Titin, coded by the *TTN* gene, is a critically important protein for myofibril elasticity and structural integrity of the sarcomere, whose splicing is altered in DM1. Its splicing is regulated by *MBNL1* but not CELF1 and work in RBFOX2 knock-out animal models suggests that RBFOX2 is also involved in *TTN* splicing. In DM1 heart samples, exons Zr4, Zr5 and Mex 5 are included in the titin transcript, coding for the foetal isoform of the protein. In this case, the splicing alteration is unique to DM1 as it has not been described in any other muscle pathology nor in healthy adult tissues. *TTN* aberrant splicing causes a defective myofibril assembly and function [52,86,113]. 

The splicing of *RBFOX2* itself is also altered in DM1 cardiac muscle samples. In this case, the isoform presented in cardiac muscle is three nucleotides shorter than usual, and this codes for the non-muscle isoform of the protein. The overexpression of this non-muscle isoform in DM1 human heart tissues is caused by a combination of elevated CELF1 and reduced miRNAs activity. It is directly correlated with the production of a pathogenic ion channel splice variants that may contribute to DM1-related cardiac conduction delay and arrhythmogenesis. This expression change hasn’t been described in any other cardiac pathology studied although it hasn’t been tested in other neuromuscular disorders [113].

### 3.2. Misregulation of mRNA Localization and Stability

Alternative splicing regulation is MBNL and CELF1′s main function, but they also take part in other cellular processes such as regulation of mRNA stability and protein decay which also might be involved in loss of cell function and viability in DM1 [33]. Interestingly, the own MBNL and CELF intracellular locations and stabilization play a key role in the misregulation of different processes. 

Although MBNLs are mainly localized in the nucleus, they are also present in the cytoplasm, albeit at a lower concentration. As described in mice, nuclear MBNLs repress or activate splicing depending on the binding location and, in the cytoplasm, MBNL binding in 3′UTRs may facilitate targeting of mRNAs with signal sequences to the rough endoplasmic reticulum. Similarly, transcripts with 3′UTR MBNL binding may be targeted to membrane-rich organelles to localized mRNA translation or driven to a mRNA isoform-specific location [33], as well as destabilized for degradation through the same 3′UTR binding. Although the exact mechanisms are unknown, it is hypothesized that mRNA localization and stability might be altered in DM1 due to lack of MBNL in the cytoplasm [33,114], supported by the suggested effects of *MBNL1* deubiquitination and delocalization in the DM1 brain [115]. 

Recently, CELF1 location has been studied in mice models, suggesting a stronger role in skeletal muscle wasting for nuclear CELF1 functions as compared to cytoplasmic CELF1 functions [116]. Mice with overexpressed nuclear CELF1 were characterized by stronger histopathological defects, muscle loss within 10 days and several changes in alternative splicing, while mice overexpressing CELF1 in the cytoplasm display changes in protein levels of targets known to be regulated at the level of translation by CELF1, but with minimal changes in alternative splicing [116]. Cytoplasmic CELF1 overexpression, as described in DM1, contributes to mRNA stability misregulation through alterations in mRNA target decay and in the regulation of protein secretory pathways. CELF1 binds to the mRNAs coding for signal recognition particle protein (SRP) subunits and promotes their decay, disabling its translation and functionality. Also, CELF1 overexpression contributes to the faster turnover of the mRNAs coding for the cytoplasmic ribonucleoprotein complex, which is a protein structure that regulates the translation of secreted and membrane-associated proteins [34,117,118]. 

MBNL trapping among *DMPK* transcripts produces RNA toxicity not only by alternative splicing of different mRNAs, but also by routes not directly associated with MBNL and CELF1 function. Nuclear speckles are non-membrane bound nuclear assemblies of macromolecules, such as splicing factors, where some pre-mRNAs are processed before being exported. Ribonuclear foci are dynamic structures that can co-localize at the periphery of these nuclear speckles. The physical interaction between ribonuclear foci and nuclear speckles may prevent the entry of other RNAs into them, altering the transcription, splicing and post-transcriptional modifications of different transcripts as a secondary effect. Many mislocalized mRNAs encode for secreted proteins, extracellular matrix components and proteins involved in cell communication. Although specific pathways haven’t been described yet, these alterations could affect proper neuromuscular junction formation [119].

### 3.3. Misregulation of mRNA Translation

CELF1 is physiologically involved in the regulation of mRNA translation and, as previously mentioned, its cytoplasmic overexpression alters normal regulation. Mature mRNAs are translated to protein in the ribosomes, mainly located in the endoplasmic reticulum in the cytoplasm. Cytoplasmic phosphorylated CELF1 interacts with the initiating factor eIF2 and induces the recruitment of the translational machinery to target mRNAs. Highly phosphorylated CELF1 proteins may increase the interactions with eIF2 and enhance protein translation of target mRNAs [69,120]. Also, CELF1 is implicated in cell cycle control during myoblast-to-myotube differentiation through binding to AKT. In DM1, CELF1 overexpression increases CELF1-AKT interactions, which leads to cyclin/CDK alterations that reduce the ability to withdraw from cell cycle during myoblast differentiation [68,69]. 

In addition to MBNL and CELF1 implications in different features of DM1 pathology, some processes might be altered independently of MBNL-CELF1 alterations or as side effects of other pathways perturbations. 

mRNA translation may be also affected in DM1 due to microRNA deregulation, as it has been described an impairment in downstream targets expression of altered microRNAs. Many microRNAs are expressed in a tissue-specific manner and show different expression patterns in development and disease [121]. In DM1, a group of microRNAs known as myomiRs (miR-1, miR-133a/b and miR-206) as well as other microRNAs, have been extensively studied in order to create a DM1 microRNA deregulation profile to help understand DM1 pathological features or to be used as disease biomarkers or therapeutic targets [122,123]. Indeed, specific microRNAs are detected in peripheral blood plasma of DM1 patients which inversely correlate with skeletal muscle strength, and they have been proposed as non-invasive biomarkers of the disease, even though further studies are needed [124].

Lastly, DM1 patient cells also show an aberrant Repeat Associated Non-AUG (RAN) translation, where unconventional translation of repeats in multiple reading frames occur, producing a repetitive peptide coding for polyglutamine that aggregate both in the nucleus and cytoplasm. These aggregates drive DM1 cells to apoptosis, indicating a possible role in DM1 pathogenicity. Indeed, the size of the repeats increases the efficiency of RAN translations, enhancing the production of toxic RNA protein, which may correlate with more severe phenotypes. Nevertheless, more studies are needed for a better understanding of protein toxicity mechanisms and the extent of it in different tissues, as it has only been described in [125,126]. 

## 4. Conclusions

Several splicing machinery errors and transcriptional alterations have been described in different tissues of DM1 patients and have been directly correlated with disease symptoms. As we report in this review, the discovery of specific gene deregulation underlying different DM1-associated phenotypes has increased in the last years but remains poorly understood. Identifying new genes and pathways altered in DM1 pathogenesis should allow us to define a more suitable description of the natural history of the disease but also find new targets for potential treatments. Nevertheless, not all splicing alterations are reproduced in all tissues even of the same patient, hampering the characterization of the splicing changes. It would be interesting to consider the publication of studies where splicing changes are not reproduced in different tissues with the aim of developing accurate endpoints for in vitro studies. 

Indeed, the large number of already described transcripts and proteins implicated difficulties in the biomarker selection for initial drug testing and assessment, as well as for in vivo trials. While no disease-modifying therapies have yet been identified, some RNA-mediated therapies like antisense oligonucleotides (AOs) have been applied to target repeats in mRNA transcripts to restore MBNL protein function in DM1. This approach has shown promising results in DM1 preclinical studies. Even though some difficulties need to be tackled as delivery issues with active compounds in vivo or the definition of target tissues due to the big variability of systems and symptoms involved. Different strategies targeting DM1 pathology in vivo have been described with different purposes; the degradation of specifically (CUG)n expanded *DMPK* transcripts, the reduction of MBNLs binding affinity to the CUG repeats and the reduction of CELF1 concentration. All of these approaches represent plausible strategies for targeting DM1 pathological implications at different levels. However, the efficacy assessment of the different compounds has been different in all cases. In some studies, they describe the reduction of ribonuclear foci formation; others focus on the reversion of some splicing aberrations at RNA levels, the depletion of *DMPK* transcripts or the redistribution of *MBNL1* protein. 

DM1 is a very complex disease with no clear endpoints for drug evaluation at pre-clinical level, nor in clinical trials. The reduction of ribonuclear foci formation could represent an interesting strategy, although foci counts are variable among cell types (fibroblasts or myoblasts) and might even present differences between myoblasts and myotubes, which emphasizes the need for a supporting measure [127]. Reversion of splicing defects might be more reproducible, as it can be quantified and normalised between cell types, groups and techniques but this requires further studies to determine an accurate representation of the cellular context or disease phenotype that could be standardized between different research groups. A gold standard evaluation at different levels, as RNA and protein, might be useful for a more successful screening of DM1 therapeutic strategies.

## Figures and Tables

**Figure 1 genes-11-01109-f001:**
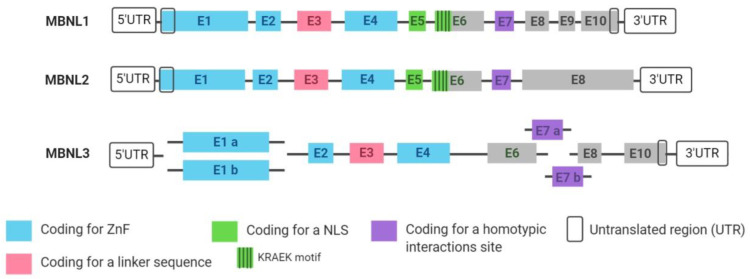
*MBNL1, MBNL2* and *MBNL3* paralog structures. All the exons that form the three MBNL paralogs are represented, each with its corresponding functional domains highlighted. Some of them are constitutively expressed in all transcripts but others undergone alternative splicing depending on the tissue or developmental process. ZnF = zinc finger; NLS = nuclear localization signal. Created with Biorender.com.

**Figure 2 genes-11-01109-f002:**
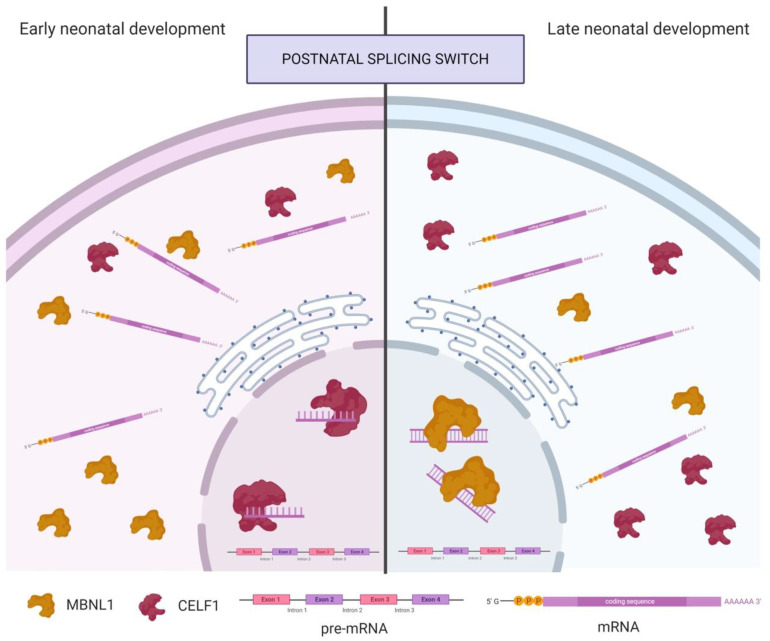
*MBNL1* and CELF1 localization in neonatal development in skeletal muscle. CELF1 is the main splicing regulator in the nucleus during early neonatal development. It binds with higher affinity to single-stranded pre-mRNAs that will later pass to the endoplasmic reticulum, where mature mRNAs will be formed. *MBNL1* is the main splicing component in the cytoplasm until the postnatal splicing switch when it is translocated to the nucleus, substituting CELF1 in the pre-mRNA binding, although in this case, with higher affinity to double-stranded sequences. Created with Biorender.com.

**Figure 3 genes-11-01109-f003:**
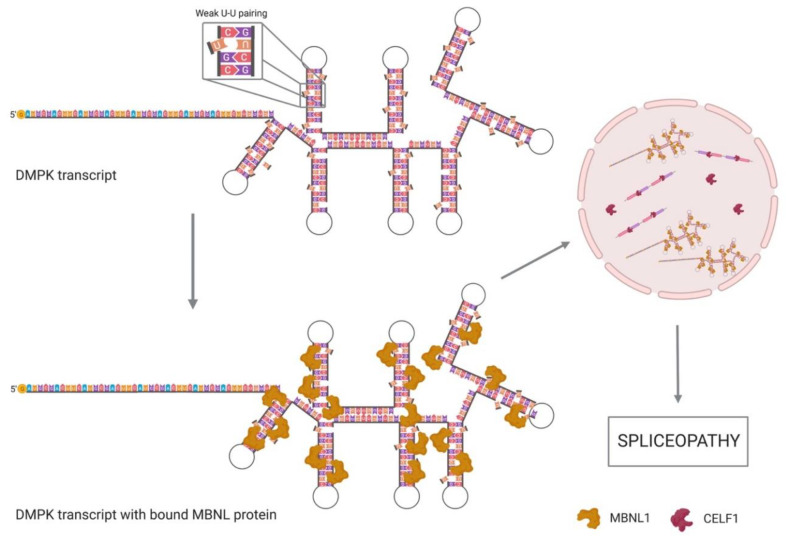
*DMPK* transcripts with CUG expansion and MBNL bound. *MBNL1* proteins are attracted to the *DMPK* transcripts CUG expansion due to the weak pairing of U–U, resulting in MBNL depletion from the nucleoplasm and lack of functional *DMPK* in the cytoplasm. CELF1 is overexpressed in DM1 tissues and regulates most of the splicing processes undergone in the nucleus, which causes the expression of embryonic variants that lead to a spliceopathy. Created with Biorender.com.

**Figure 4 genes-11-01109-f004:**
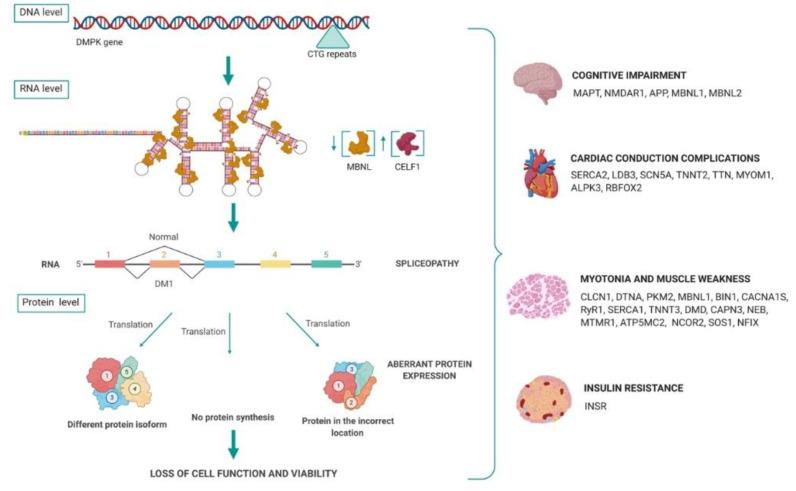
DM1 spliceopathy overview. At DNA level, CTG repeats code for hairpin CUG structures that bind *MBNL1* proteins with high affinity. Due to *MBNL1* loss, CELF1 is overexpressed through PKC phosphorylation and alters the splicing of different transcripts, mainly switching to embryonic isoforms. The splicing deregulation or spliceopathy induces an aberrant protein expression that provokes the loss of cell function and viability. Different transcripts from different tissues are incorrectly spliced, causing most DM1 symptoms. Created with Biorender.com.

**Table 1 genes-11-01109-t001:** Summary of transcripts expressed in brain tissue and altered in human DM1 samples. All genes mentioned are described in the text and represented in alphabetical order. DM1 patients brain sections refer to extraction of total RNA from homogenised brain tissue from DM1 patient autopsy, DM1 patient skeletal muscle biopsy refers to extraction of total RNA from homogenised skeletal muscle biopsy of DM1 patients and DM1 patient-derived myotubes refers to RNA extraction of cultured myoblasts from DM1 patient biopsy differentiated to myotubes in cell culture.

Gene	Splicing Alteration		Reference	Tissue Expression	Implications in DM1 Pathology	Sample Type
	Exon/Intron	Inclusion/Exclusion				
***APP***	Exon 7	Exclusion	Jiang (2004) [85]	Brain	n.d.	DM1 patients brain sections
***MAPT***	Exon 2	Exclusion	Goodwin (2015) [83]	Brain (frontal cortex)	Progressive appearance of NFTs composed of intraneuronal aggregates of hyperphosphorylated tau protein.	DM1 patients brain sections
	Exon 3	Exclusion				
	Exon 10	Exclusion				
***MBNL1***	Exon 6	Inclusion	Dhaenens (2008) [87]	Most tissues	Splicing defects	DM1 patients brain sections
	Exon 8	Inclusion				
***MBNL2***	Exon 7	Inclusion	Nakamori (2013) [29]	Brain	Splicing defects	DM1 patient-derived cultured myotubes
	Exon 8	Inclusion	Yamashita (2012) [86]			
***NMDAR1***	Exon 5	Inclusion	Jiang (2004) [85]	Brain	Memory impairment.	DM1 patients brain sections

**Table 2 genes-11-01109-t002:** Summary of transcripts expressed in skeletal muscle altered in DM1 samples. All genes mentioned are described in the text and represented in alphabetical order. DM1 patient skeletal muscle biopsy refers to extraction of total RNA from homogenised skeletal muscle biopsy of DM1 patients and DM1 patient-derived myotubes refers to RNA extraction of cultured myoblasts from DM1 patient biopsy differentiated to myotubes in cell culture.

Gene	Splicing Alteration		Reference	Tissue Expression	Implications in DM1 Pathology	Sample Type
	Exon/Intron	Inclusion/Exclusion				
***ALPK3***	Exon 2	Inclusion	Nakamori (2013) [29]	Cardiac muscle	Not described	DM1 patient skeletal muscle biopsy
***ATP2A1***	Exon 22	Exclusion	Kimura (2005) [98]	Skeletal muscle	Muscle degeneration: impair intracellular calcium homeostasis	DM1 patient-derived cultured myotubes
***ATP5MC2***	Exon 1	Inclusion	Yamashita (2012) [86]	Skeletal muscle	n.d.	DM1 patient skeletal muscle biopsy
***BIN1***	Exon 11	Exclusion	Fugier (2011) [96]	Skeletal muscle	Muscle weakness: altered excitation–contraction coupling	DM1 patient skeletal muscle biopsy
***CACNA1S***	Exon 29	Exclusion	Tang (2012) [97]	Skeletal muscle	Muscle weakness: altered excitation–contraction coupling	DM1 patient skeletal muscle biopsy
***CAPN3***	Exon 16	Exclusion	Yamashita (2012) [86]	Skeletal muscle	Muscle weakness: decreased protease activity	DM1 patient skeletal muscle biopsy
***CLCN1***	Intron 2	Inclusion	Charlet (2002) [93] Mankodi (2002) [94]	Skeletal muscle Brain	Myotonia	DM1 patient skeletal muscle biopsy
	Exon 7a	Inclusion	Lueck (2006) [106] Nakamori (2013) [29]			
***DMD***	Exon 71	Exclusion	Yamashita (2012) [86]	Skeletal muscle	Muscle weakness: alteration of the membrane integrity	DM1 patient-derived cultured myotubes
	Exon 78	Exclusion	Rau (2015) [107]			
***DTNA***	Exon 11a	Exclusion	Nakamori (2013) [29]	Brain Cardiac muscle Skeletal muscle	Muscle weakness	DM1 patient skeletal muscle biopsy
	Exon 12	Exclusion				
***FHOD***	Exon 11a	Exclusion	Yamashita (2012) [86]	Skeletal muscle	n.d.	DM1 patient skeletal muscle biopsy
***GFPT1***	Exon 9	Exclusion	Nakamori (2013) [29]	Most tissues	n.d.	DM1 patient skeletal muscle biopsy
***INSR***	Exon 11	Exclusion	Savkur (2001) [103]	Skeletal muscle	Insulin resistance: decreased metabolic response to insulin	DM1 patient skeletal muscle biopsy and cultured myotubes derived from DM1 patient fibroblasts
***MBNL1***	Exon 5	Inclusion	Konieczny (2014) [55]	Skeletal muscle	Splicing defects	DM1 patient skeletal muscle biopsy
	Exon 7	Inclusion	Nakamori (2013) [29]			
	Exon 10	Inclusion	Yamashita (2012) [86]			
***MYOM1***	Exon 17a	Inclusion	Koebis (2011) [101]	Skeletal muscle	Sarcomeric M-band instability	DM1 patient skeletal muscle biopsy
***MTMR1***	Exon 2.1	Inclusion	Buj-Bello (2002) [104]	Skeletal muscle	Impaired myogenesis	DM1 patient-derived cultured myotubes
	Exon 2.2	Inclusion	Yamashita (2012) [86]			
	Exon 2.3	Inclusion	Buj-Bello (2002) [104]			
***MXRA7***	Exon 4	Exclusion	Yamashita (2012) [86]	Most tissues	n.d.	DM1 patient skeletal muscle biopsy
***NCOR2***	Exon 10	Inclusion	Yamashita (2012) [86]	Skeletal muscle	n.d.	DM1 patient skeletal muscle biopsy
***NEB***	Exon 116	Inclusion	Yamashita (2012) [86]	Skeletal muscle	n.d.	DM1 patient skeletal muscle biopsy
***NFIX***	Exon 7	Inclusion	Yamashita (2012) [86]	Skeletal muscle	n.d.	DM1 patient skeletal muscle biopsy
***NRAP***	Exon 12	Exclusion	Lin (2006) [52]	Skeletal muscle	Altered myofibril assembly.	DM1 patient skeletal muscle biopsy
***PKM2***	Exon 9	Inclusion	Gao (2013) [89]	Brain Type I fibres from skeletal muscle in DM1	Defects in energy metabolism in skeletal muscle	DM1 patient skeletal muscle biopsy
	Exon 10	Inclusion				
***RYR1***	Exon 70	Exclusion	Kimura (2005) [98]	Skeletal muscle	Muscle weakness: decreased muscle contraction	DM1 patient-derived cultured myotubes
***SMYD1***	Exon 39	Inclusion	Du (2010) [50]	Skeletal muscle Cardiac muscle	n.d.	DM1 patient skeletal muscle biopsy
***SOS1***	Exon 25	Exclusion	Yamashita (2012) [86]	Skeletal muscle	Inhibits signalling pathways involved in muscle hypertrophy	DM1 patient skeletal muscle biopsy
***TNNT3***	Exon 23	Inclusion	Yamashita (2012) [86]	Skeletal muscle	Alteration of the sarcomere structure	DM1 patient skeletal muscle biopsy

**Table 3 genes-11-01109-t003:** Summary of transcripts expressed in cardiac tissue altered in human DM1 samples. All genes mentioned are described in the text and represented in alphabetical order. DM1 patient skeletal muscle biopsy and DM1 patient cardiac muscle tissue refers to extraction of total RNA from homogenised skeletal muscle biopsy or cardiac muscle autopsy, respectively, of DM1 patients.

Gene	Splicing Alteration		Reference	Tissue Expression	Implications in DM1 Pathology	Sample Type
	Exon/Intron	Inclusion/Exclusion				
***ATP2A2***	Intron 19	Inclusion	Kimura (2005) [98] Dixon (2015) [111]	Cardiac muscle	Cardiac conduction impairment: deregulated calcium influx	DM1 patient-derived cultured myotubes
***RBFOX2***	3 nt	Exclusion	Misra (2020) [113]	Cardiac muscle	Cardiac conduction delay and arrhythmogenesis	DM1 patient cardiac muscle tissue
***SCN5A***	Exon 6a	Inclusion	Freyermuth (2016) [110]	Cardiac muscle	Conduction slowing: decreased upstroke of the cardiac action potential	DM1 patient cardiac muscle tissue
***TNNT2***	Exon 5	Inclusion	Ho (2004) [54] Dixon (2015) [111]	Cardiac muscle	Alteration of the contractile properties: different calcium sensitivity of the myofilament	DM1 patient skeletal muscle biopsy
***TTN***	Zr4	Inclusion	Lin (2006) [52] Yamashita (2012) [86]	Cardiac muscle	Defective myofibril assembly and function	DM1 patient skeletal muscle biopsy
	Zr5	Inclusion				
	Mex5	Inclusion				
***ZASP/LDB***	Exon 5	Inclusion	Nakamori (2013) [29]	Cardiac muscle	Morphological abnormalities of the cardiac fibre	DM1 patient skeletal muscle biopsy
	Exon 11	Inclusion				

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
