# Peer review of "An Overview of Alternative Splicing Defects Implicated in Myotonic Dystrophy Type I"

_genes, 2020, doi:10.3390/genes11091109_

Round 1

Reviewer 1 Report

The Review manuscript by López-Martinez et al. describes the molecular basis of myotonic dystrophy type 1 pathomechanism. It is a comprehensively written manuscript that requires some changes. First, the text, including the abstract, needs some integration to become more coherent. Now it seems a bit chopped. Second, many parts of the manuscript are too speculative, controversial, or incorrect and should be modified. They are marked with << >>. Third, I suggest reviewing recent literature (last 3-5 years) to fill the gaps and update the text.

Major comments:

"Within expanded DMPK transcripts, CUG repeats form imperfect stable <<short>> hairpin structures that accumulate in the cell nucleus in small ribonuclear complexes or microscopically visible inclusions, which bind (or sequester) proteins implicated in <<transcription>>, splicing or <<RNA export>>." Please, provide evidence for this statement.

"More than <<thirty>> splicing events are mis-regulated due to MBNL1 sequestration and CELF1 upregulation" and "To date, more than thirty transcripts have been found to be misspliced in different tissues in DM1 patients (Figure 4) and more than sixty in mice tissues [46,69]." It is way more than 30 alerted splicing events in DM, most likely a few hundred. See: PMID: 23929620, PMID: 30561649, PMID: 28698297.

<<Figure 1>> MBNL3 does not encode exon 5 coding NLS.

"The three paralogs are composed by <<13 exons>> in total, some of which are alternatively spliced in the different transcript variants of each paralog." Inconsistent with Figure 1.

"Exon 3 encodes for a linker sequence essential for RNA/protein interactions as <<it increases protein flexibility, allowing the binding to a wide range of targets with different structures>>." Speculation. MBNL exon 3 impacts protein activity. Provide evidence showing it plays a role in target selection?

"MBNL1 transcript variants with exons 5 and 7 are mainly described in early differentiation stages and in adult DM1 tissues. <<These exons enhance sequestration>> of MBNLs in nuclei of DM1 cells and <<thus contribute to the severity of the phenotype>>." Too simplified. Please provide evidence for this statement.

"Studies in mice suggest that <<MBNL1 has a predominantly cytoplasmic location during early stages of neonatal development>>, while at the end of the process is predominantly nuclear>>, coinciding with the post-natal splicing transitions (Figure 2). Therefore, <<postnatal splicing transitions are triggered, at least partly, by translocation of MBNL1 from the cytoplasm to the nucleus>>." Although Lin et., 2006 is a crucial early paper, Fig. 6 can be misleading. Other articles should also be invoked here, for example, PMID: 19075228. Figure 2 should be revised.

"MBNL <<stabilizes the hairpin structure through the intercalation between complementary strands>>, as represented in Figure 3." Please provide evidence for this statement.

"The splicing defects described in DM1 are strikingly similar to those observed in Mbnl1 knockout mice <<but not in Mbnl2 defective mice>>, leading to the conclusion that <<MBNL1 has a pivotal role in DM1 pathogenesis, independently of MBNL2>>, also "MBNL1 sequestration is enough to explain misregulated splicing in adult DM1 skeletal muscle, so just one splicing factor missing is responsible of most downstream effects." In most cases, the depletion of MBNL paralogs has an addictive effect on spliceopathy that varies in different tissues. See PMID: 22884328, PMID: 24293317, PMID: 28698297.

"Although with lower affinity than MBNL1, <<CELF1 binds to CUG-repeat containing RNA, leading to the activation of the protein kinase C (PKC) signalling pathway>> that promotes CELF1 hyperphosphorylation and stabilization, increasing its steady-state levels[59–61]." Please provide evidence for this statement. 

"Figure 4 summarises some DM1 symptoms <<linked>> to the splicing alteration of different genes in pathogenic conditions, while transcript alterations of the genes represented are detailed later in this manuscript". Some of the indicated transcripts are not proven to cause DM1 symptoms. For example, how NCOR2 alternative 5' splice site selection in DM1 impacts myotonia or muscle weakness? Also, some descriptions on pages 11-22 do not provide real insight into DM1 pathogenesis. I suggest reducing this section to the relevant splicing events like BIN1 or CACNA1S.

"The inclusion of exons 7 and 8 in MBNL2 and exon <<6>> and 8 in MBNL1 (recapitulating MBNL3) mRNAs, has been described as an enhancer of alternative splicing deregulation in DM1 patients [9,76,77]". Number inconsistent with Figure 1. All MBNL alternative exons have a potential impact on splicing. See: PMID: 27903900, PMID: 27733504.

"This leads to the misregulation of alternative splicing of multiple transcripts represented in Table 2, included MBNL1 itself: exon 7 is developmentally regulated by MBNL1 homotypic interactions during postnatal development[9,45,76] and it has been described the inclusion of exons 5, 7 and <<10>> of the MBNL1 transcript in DM1 skeletal muscle 299 samples." Exon 10 is 3'UTR, according to Figure 1.

"MBNL1 own splicing alterations affect other gens by generating different protein isoforms, ablating protein synthesis, or changing protein localization, such as ectopic expression of proteins as dystrobrevin α and pyruvate kinase M2." Provide references. 

"This binding induces the destabilization of the protein and activates degradative pathways such as ubiquitination." Unclear sentence. 

The sections about 3.2 and 3.3 are too superficial and not well integrated.

Minor/editorial comments:

-Homogenize nomenclature: myotonic dystrophy type I or 1 (title vs. abstract vs. main text); Fig (line: 210)

-Develop the acronym when used for the first time: 3'UTR (lines 15 and 45).

-Inconsistent italicization, bolding and underlining: DMPK (line 16), spliceopathy (line 21), more on p.8+.

-Remove or add extra spaces/symbol (lines 27, 121, 191, 198, 272, 482, 569).

-Consider removing the word "regular" (line 228), changing "technique" to more appropriate term (line 617).

Reviewer 2 Report

López-Martinez et al. present a review of the spliceopathy associated with myotonic dystrophy type I. The authors cover muscleblind proteins, CELF1, pathogenesis of DM1 and then list the genes that have been demonstrated to exhibit DM1-associated alternative splicing patterns in brain, skeletal muscle and heart. Overall, the opinion of this reviewer is positive. The review is timely and will likely serve as a useful reference for others in this field. The manuscript also contains some high quality figures. The manuscript contains multiple typographical/English language issues. I have highlighted some of these below, although some editing is recommended to fix all of these errors.

Main Points

  1. As far as I can tell, the authors do not ever discuss the possibility that certain specific alternative splicing events observed in DM1 tissues are co-incidental and do not contribute substantially to disease pathology. (i.e. not all splice changes are pathogenic).
  2. I think that the authors should also discuss the following study:

Neuromuscul Disord. 2014 Mar; 24(3): 227–240.

Most expression and splicing changes in myotonic dystrophy type 1 and type 2 skeletal muscle are shared with other muscular dystrophies

Linda L. Bachinski,a Keith A. Baggerly,b Valerie L. Neubauer,a Tamara J. Nixon,a Olayinka Raheem,c Mario Sirito,a Anna K. Unruh,b Jiexin Zhang,b Lalitha Nagarajan,a Lubov T. Timchenko,d Guillaume Bassez,e Bruno Eymard,f Josep Gamez,g Tetsuo Ashizawa,h Jerry R. Mendell,i Bjarne Udd,b,j,k and Ralf Krahea,l,m,*

  1. In the abstract the authors say:

‘due to a foetal-to-adult splicing switch’

I believe this is a mistake, and it is clear from the rest of the manuscript that the authors mean:

‘adult-to-foetal’

  1. In Figure 3, MBNL and CELF1 proteins need to be labelled on the figure.

  1. In the text and in Figure 3 the following point is made:

‘MBNL1 proteins are attracted to the DMPK transcripts CUG expansion due to the weak pairing of U-U and A-A bases’

What is the relevance of A-A bases here? In DM1 the DMPK mRNA repeat consists only of CUG.

  1. Section 3.2 is vague and would benefit from more specific examples, and citations.

  1. The authors state:

‘Indeed, specific microRNAs are detected in peripheral blood plasma of DM1 patients which inversely correlate with skeletal muscle strength, and they have been proposed as non-invasive biomarkers of the disease, even though further studies are needed [107–109].’

I do not believe this has been appropriately cited. Only one of these citations refers to plasma miRNAs. The authors should amend and check that all citations throughout the manuscript are appropriate.

  1. Citations should be added to tables (not just first author name and date).

  1. Conventions for gene naming should be observed consistently throughout. Official gene symbols should be italicised when referring to the gene or mRNA product. Use of official gene symbols is preferable although non-standard, but commonly-used, symbols are fine provided the official gene symbol is shown in parentheses following the first usage. For example, RyR1 (RYR1).

  1. I don’t agree with following point:

‘RBPs are specific for each differentiation status and contribute to splicing coordination. In fact, genes regulated by alternative splicing are not modulated at their overall expression levels (they are up-and down-regulated) [22,26].’

I think this is an over-generalisation. It is not difficult to think of examples whereby alternative splicing could lead to changes in transcript level. Indeed, Stoke Therapeutics is utilising alternative splicing as a means of protein up-regulation. Also consider non-sense-mediated decay. At the least, the language needs to be toned down.

Minor Points

  1. ‘of cellular mRNAs translation’ change to: ‘of cellular mRNA translation’
  2. ‘as the latest are much larger’ change to: ‘as the latter are much larger’
  3. ‘as 38.268 genes there are as many as 109.005 mRNAs’ change to: ‘as 38,268 genes there are as many as 109,005 mRNAs’
  4. ‘Its splicing program is so complex that might differ even to cardiac muscle’s program.’ Change to: ‘Its splicing program is so complex that it might differ even to cardiac muscle’s program.’
  5. ‘MBNL3, which expression is more restricted’ change to: 5. ‘MBNL3, whose expression is more restricted’
  6. ‘are composed by’ change to: ‘are composed of’
  7. ‘but not two double-stranded sequences’ change to: ‘but not to double-stranded sequences’ (if I understand the meaning correctly.
  8. ‘Embrionary’ change to: ‘embryonic’
  9. ‘is directly related with the length of expansion’ change to: ‘is directly related to the length of expansion’
  10. ‘as DMPK mRNA is withhold in’ change to: ‘as DMPK mRNA is withheld in’
  11. ‘affect other gens by generating’ change to: ‘affect other genes by generating’
  12. In the following sentence:

‘Different CAPN3 mutations have been described causing LGMDR1 but any similar to CAPN3 transcript alterations in DM1’

The intended meaning is not clear to me.

Round 2

Reviewer 1 Report

López-Martinez et al. significantly improved the text and figure in the revised manuscript, but few omissions left.

Major comments:

1. "Thank you for the point. The cited sentence is mentioned in Tran, 2011 (PMID: 21454535) and reviewed in Konieczny, 2014 (PMID: 25183524), both referenced at the end of the paragraph".

Konieczny, 2014 (PMID: 25183524) is a review paper, and Tran, 2011 (PMID: 21454535) demonstrates MBNL1 with exon7 homotypic aggregation without repeats (Fig. 7). I suggest citing PMID: 17702765 (see Figs. 5 and 6).

2. "Thank you very much for the observation, the references (PMID: 16533306 and PMID: 25183524) were missing, and they have been properly added".

Both PMID: 16533306 and PMID: 25183524 are review papers. Please, confront this sentence with recent structural articles, such as PMID: 27720642, PMID: 29955876, PMID: 30700578, etc.

3. Please correct "<<short-hairpin>> CUG structures" in Figure 4 legend. 

4. I agree with the Authors. Nakamori et al. developed biomarkers associating with DM muscle phenotype, and it could be more clearly reflected in the text. For example, a sentence "Different transcripts from different tissues are incorrectly spliced, causing 243 most of DM1 symptoms", suggests a direct link between phenotype and transcripts listed in Figure 4.

5. “In adult and DM1 brain tissue, different MBNL1 and MBNL2 isoforms are expressed, with an increase of, isoforms including exons 7 and 8 in MBNL2 mRNA and exon <<6>> and 8 in MBNL1 mRNA [29,84,85]”.

MBNL2 exon 6 is not alternatively spliced.

Minor comments:

Typos: "muscle fibres" (lines 144 and 160), "signalling" (lines: 216, 419, 465, 470, 484, Table 2).
